# Integrated Transcriptomics and Metabolomics Analysis Reveals Convergent and Divergent Key Molecular Networks of Dominant Genic Male Sterility and Cytoplasmic Male Sterility in Cabbage

**DOI:** 10.3390/ijms26031259

**Published:** 2025-01-31

**Authors:** Nan Zhang, Linqian Kuang, Limei Yang, Yong Wang, Fengqing Han, Yangyong Zhang, Shaohui Wang, Honghao Lv, Jialei Ji

**Affiliations:** 1College of Plant Science and Technology, Beijing University of Agriculture, Beijing 102206, China; zhangnan620421@163.com (N.Z.); wangshaohui@bua.edu.cn (S.W.); 2State Key Laboratory of Vegetable Biobreeding, Institute of Vegetables and Flowers, Chinese Academy of Agricultural Sciences, Beijing 100081, China; klq964136556@163.com (L.K.); yanglimei@caas.cn (L.Y.); wangyong@caas.cn (Y.W.); hanfengqing@cass.cn (F.H.); zhangyangyong@caas.cn (Y.Z.); 3Collage of Horticulture, Hunan Agricultural University, Changsha 410128, China

**Keywords:** Ogura CMS, DGMS, pollen development, cabbage

## Abstract

Cytoplasmic male sterility (CMS) and dominant genic male sterility (DGMS) both result in the inability to produce or release functional pollen, making them pivotal systems in the hybridization breeding programs of *Brassica* crops such as cabbage (*B. oleracea* var. *capitata*). However, the underling molecular mechanisms are still largely unexplored. This study integrated transcriptomic and metabolomic analyses of cabbage DGMS line, Ogura CMS line, and the maintainer line to uncover the molecular mechanisms underlying these sterility types. The joint analysis predominantly identified significantly enriched pathways, including carbohydrate metabolism, flavonoid biosynthesis, and phenylpropanoid pathways between the MS lines and the maintainer. Especially, the CMS line exhibited a broader range of metabolic perturbations, with a total of 3556 significantly differentially expressed genes (DEGs) and 439 differentially accumulated metabolites (DAMs) detected, particularly in the vitamin B6 metabolism pathway, which showed significant alterations. Given the differences in the inactivation period of microspores in CMS and DGMS lines, we found that DEGs unique to DGMS and maintainer line, such as BoGRPs and BoLTPs, primarily regulate fertility development before the unicellular stage. The DEGs shared between CMS_vs_maintainer and DGMS_vs_maintainer mainly govern microspore development after release from the tetrad, such as BoHXK1 and BoIDH. Additionally, the DEGs unique to CMS_vs_maintainer may contribute to other damage in floral organs beyond male fertility, potentially leading to severe bud abortion, such as BoPNPO. These findings provide a comprehensive framework for understanding the molecular mechanisms of male sterility and offer valuable insights into future breeding strategies in cruciferous vegetables.

## 1. Introduction

Cabbage (*Brassica oleracea* L. var. *capitata*) is one of the most widely cultivated vegetable varieties worldwide [1,2], belonging to the Brassica genus in the Cruciferae family [3,4]. The global sown area of cabbage in 2023 was 2.36 million hm^2^ (Food and Agriculture Organization of the United Nations data). Cabbage is a typical dioecious crop with significant hybrid vigor [5,6,7,8]. Currently, most cabbage varieties globally are hybrids. The use of male sterility lines for hybrid vigor breeding is now the primary method for developing new cabbage varieties [9,10,11]. The main techniques for producing cabbage F1 hybrids include cytoplasmic male sterility and dominant male sterility systems [12,13,14,15,16].

Cytoplasmic male sterility (CMS) is the most widely used male sterility type, characterized by poor flower bud development, a high incidence of bud abortion, and complete male sterility. While CMS allows for easy hybrid breeding and thorough sterility, it is also associated with a high occurrence of dead flower buds, resulting in reduced hybrid seed quality and yield, as well as decreased genetic diversity and increased susceptibility to diseases [17,18,19]. In contrast, cabbage dominant gene male sterility (DGMS) lines exhibit fewer dead buds, larger flowers, and well-developed nectar glands, leading to higher seed yields and quality when used as female parents in hybrid production. However, the hybridization process is more complex and influenced by the genetic background of the breeding material. Furthermore, DGMS lines are sensitive to low temperatures, although they exhibit fewer dead flower buds, thus offering broad application prospects [20,21,22,23,24,25]. Although both types of male sterility are widely used in cabbage breeding, the precise regulatory networks and their connections to metabolic changes remain poorly understood, necessitating further exploration to fully elucidate the molecular basis of sterility.

In this study, we investigated changes in bud metabolite and gene expression profiles across the DGMS and CMS lines through whole-transcriptome and non-targeted metabolomics. Transcripts and metabolites specifically enriched in Ogura CMS and DGMS were identified, and the regulatory networks of co-enriched and differentially enriched transcripts and metabolites were mapped for these lines. These findings establish a robust foundation for future research on pollen development and the regulatory mechanisms of Ogura CMS and DGMS. Furthermore, this study sought to identify the key molecular networks regulating the early and late stages of male fertility in cabbage, offering theoretical and technical support for hybrid breeding strategies based on male sterility.

## 2. Results

### 2.1. TEM Differences Between Ogura CMS20-2-5 and DGMS20-2-5

The transmission electron microscopy (TEM) images of pollen grains at the same developmental stage revealed that DGMS20-2-5 exhibits early antheretrichia, with an abnormal pollen wall structure. During the tetrad stage (S7), the inner and outer walls are absent, and the outer wall has an irregular structure that cannot be distinctly classified as either a columnar or covering layer. During the mononuclear pollen grain stage (S10), 20-2-5 microspores generally develop into mononuclear pollen grains, whereas DGMS20-2-5 microspores show signs of cytoplasmic degradation, including pronounced vacuolization and the presence of filaments. Ogura CMS20-2-5 displays normal tetrad development during S7 but lacks a normal outer wall at S10, resulting in developmental arrest and the gradual degeneration of pollen grains (Figure 1A,B).

### 2.2. Transcriptome Sequencing and Differential Gene Screening

This study employed full-length transcriptome sequencing of the maintainer line 20-2-5 (M1), Ogura CMS line 20-2-5 (M2), and DGMS line 20-2-5 (M3) using the Illumina NovaSeq 6000 platform (Illumina, San Diego, CA, USA). After low-quality data were filtered out and adapters were removed, high-quality clean data were obtained (Table 1). Following quality control, 62.95 Gb of clean data was generated, with the Q30 base percentage for all samples exceeding 91.61%, indicating high sequencing quality suitable for subsequent analysis. The HISAT2 [26] software was used to align the clean reads to the *Brassica* reference genome (http://brassicadb.cn/#/Download/ [27], accessed on 7 April 2024), ensuring high precision and efficiency. The read alignment information from the reference genome was extracted, and StringTie [28] was used to assemble the aligned reads and reconstruct the transcriptome for further analysis. The alignment efficiency ranged from 72.51% to 73.66% across all samples. Pairwise comparisons of M1, M2, and M3 were conducted to identify differentially expressed genes (DEGs) using the “A_vs_B” naming convention, where A represents the control group and B represents the experimental group. DEGs were selected based on a false discovery rate (FDR) of <0.01 and |log2FC| ≥ 1 (Fold Change, FC). Using M1 as the control, 1651 significant DEGs were identified in the M1_vs_M2 comparison (Appendix A), with 322 upregulated and 1329 downregulated genes. In the M1_vs_M3 comparison, 3556 significant DEGs were identified, with 1690 upregulated and 1966 downregulated genes (Appendix A). A total of 2588 significant DEGs were detected in the comparison between the DGMS and CMS lines (M2 vs. M3), with 1615 upregulated and 973 downregulated genes (Figure 2A). The number of DEGs detected in M3 was approximately twice that in M2, suggesting that more genes may be involved in regulating CMS, including transcriptional and post-transcriptional processes (Figure 2E,F). The Venn analysis (Figure 2B and Appendix A) revealed that 1341 DEGs were significantly differentially expressed in both the M1_vs_M2 and M1_vs_M3 comparisons, with 193 genes upregulated and 1136 downregulated. Twelve genes exhibited inconsistent expression patterns between M2 and M3, being downregulated in the M1_vs_M2 comparison and upregulated in the M1_vs_M3 comparison. These DEGs suggest that these genes may play a role in key fertility-regulating pathways, thereby influencing floral development. Furthermore, 2315 DEGs were uniquely expressed in M3. The accuracy of the RNA-seq results was further validated via qRT-PCR.

To explore the molecular mechanisms underlying pollen sterility associated with Ogura CMS and DGMS, we functionally annotated DEGs based on gene annotation results, classifying them into the second-level categories of the Gene Ontology (GO) database (Figure 2C,D). By comparing M1 and M2, GO analysis revealed that the biological processes associated with DGMS development primarily involve metabolic processes, cellular processes, biological regulation, and responses to stimuli. Cellular components are primarily associated with cellular anatomical entities, intracellular regions, and protein-containing complexes. Molecular functions are primarily linked to catalytic activity, binding, transporter activity, and regulating molecular functions. In contrast, the GO categories associated with CMS development in the M1_vs_M3 comparison were similar to those in the M1_vs_M2 comparison, but the number of involved genes was greater in CMS. The Cluster of Orthologous Groups (COG) classification of DEGs for both groups (Appendix A) revealed that the four most abundant categories of differential genes were carbohydrate transport and metabolism, signal transduction mechanisms, general function prediction only, and lipid transport and metabolism. Notably, in the M1_vs_M3 comparison, the number of DEGs in categories related to translation, ribosomal structure and biogenesis, posttranslational modification, protein turnover, and chaperones was three times higher than in the M1_vs_M2 comparison, suggesting that these pathways may play a unique role in the Ogura CMS 20-2-5 line.

KEGG (Kyoto Encyclopedia of Genes and Genomes) pathway analysis bubble plots (Figure 2G,H) revealed that the enriched pathways in the M1_vs_M2 comparison were primarily associated with plant–pathogen interactions, pentose and glucuronate interconversions, phosphatidylinositol signaling, and phenylpropanoid biosynthesis. In contrast, the enriched pathways in the M1_vs_M3 comparison were primarily related to plant–pathogen interactions, pentose and glucuronate interconversions, alpha-linolenic acid metabolism, phenylpropanoid biosynthesis, and ribosome biogenesis in eukaryotes. The phosphatidylinositol signaling and inositol phosphate metabolism pathways were significantly enriched only in the M1_vs_M2 comparison, while alpha-linolenic acid metabolism, ribosome biogenesis in eukaryotes, and fatty acid degradation were significantly enriched only in the M1_vs_M3 comparison.

The KEGG analysis of the 1341 DEGs significantly expressed in M2 and M3 (Appendix A) revealed that the shared enriched pathways between DGMS 20-2-5 and CMS 20-2-5 primarily include the biosynthesis of secondary metabolites, metabolic pathways, plant–pathogen interactions, starch and sucrose metabolism, and pentose and glucuronate interconversions. The pathways uniquely enriched in the DEGs expressed solely in CMS 20-2-5 include the biosynthesis of secondary metabolites, metabolic pathways, plant–pathogen interactions, plant hormone signal transduction, and starch and sucrose metabolism.

### 2.3. Metabolomic Measurements and Analysis

We performed an untargeted metabolomic analysis of three *Brassica* materials using the UPLC-MS/MS platform to compare their metabolic profiles. Initially, a correlation analysis was conducted on the samples to assess biological reproducibility within each group. The square of the Spearman rank correlation coefficient (r^2^) was used to evaluate the correlation between biological replicates. As shown in Figure 3A, the r^2^ values for all three groups exceeded 0.85, suggesting that the detected differential metabolites are reliable. This confirms that the samples exhibit good reproducibility, making them suitable for subsequent metabolomic analysis. A principal component analysis (PCA) of the three groups (Appendix A) showed that all biological replicates clustered together, further confirming the high reliability of the metabolomic data. The contributions of PC1 and PC2 were 52.64% and 31.62%, respectively, accounting for a total variance of 84.26% (Appendix A). The PCA score scatter plot showed significant differences between M1, M2, and M3, indicating that the metabolic phenotypes of these groups are distinctly differentiated.

A total of 1192 metabolites were identified across the three samples (Appendix A). Differentially accumulated metabolites (DAMs) were identified using a combination of FC, t-test *p*-values, and OPLS-DA model analysis, with the selection criteria set as FC > 1, *p*-value < 0.05, and VIP > 1. Between M1 and M2, 300 DAMs were detected (149 downregulated and 151 upregulated), and between M1 and M3, 439 DAMs were identified (237 downregulated and 202 upregulated) (Appendix A). A comparative analysis of the three sets of DAMs revealed 75 common metabolites, with 65, 93, and 84 specific metabolites identified in M1_vs_M2, M1_vs_M3, and M2_vs_M3, respectively. The overlap of differential metabolites between M1_vs_M2 and M1_vs_M3 included 191 metabolites (Figure 3B). The volcano plots further highlighted that M1_vs_M3 exhibited many differential metabolites involved in CMS regulation (Appendix A). The annotated differential metabolites were statistically analyzed using metabolite classification data from the Human Metabolome Database (HMDB), as shown in Figure 3C,D. Of the top 10 most frequently annotated pathways for both M1_vs_M2 and M1_vs_M3, several pathways were common, including carboxylic acids and derivatives, fatty acyls, organooxygen compounds, organonitrogen compounds, benzene and substituted derivatives, pyridines and derivatives, imidazopyrimidines, purine nucleosides, and cinnamic acids and derivatives. However, specific pathways unique to M1_vs_M2 included purine nucleotides, whereas M1_vs_M3 uniquely featured indoles and derivatives.

To further elucidate the functional roles of metabolites in anther development within the DGMS and CMS systems of *Brassica*, we annotated all differential metabolites using the KEGG database and selected the top 20 categories with the most annotation entries to explore the related metabolic pathways (Figure 3E,F). In the M1_vs_M2 KEGG scatter plot (Figure 3E), metabolites related to ABC transporters were the most abundant (14 metabolites), followed by cysteine and methionine metabolism (6 metabolites), glucosinolate biosynthesis (5 metabolites), linoleic acid metabolism (4 metabolites), and glycine, serine, and threonine metabolism (3 metabolites). Other pathways, such as caffeine metabolism, sulfur metabolism, glycolysis/gluconeogenesis, flavonoid biosynthesis, and biosynthesis of unsaturated fatty acids, each contributed three metabolites. In the M1_vs_M3 KEGG enrichment scatter plot (Figure 3F), the biosynthesis of the amino acid pathway was the most enriched, comprising 17 metabolites, followed by aminoacyl-tRNA biosynthesis (10 metabolites), phenylpropanoid biosynthesis (9 metabolites), and the biosynthesis of various secondary metabolites—part 2 (7 metabolites). The vitamin B6 metabolism, linoleic acid metabolism, and glycine, serine, and threonine metabolism pathways each included six metabolites, while fatty acid biosynthesis contained five metabolites; tropane, piperidine, and pyridine alkaloid biosynthesis and the biosynthesis of unsaturated fatty acids both contained four metabolites.

A clustering analysis of the selected DAMs revealed distinct features, highlighting significant differences in the metabolic profiles of the materials and reflecting the metabolic changes during anther development. A clustering heatmap was generated for the 75 common DAMs shared between M1, M2, and M3 (Figure 3G). Most DAMs were downregulated in M3 (CMS20-2-5), while metabolites such as Pterolactam, 6′-O-feruloyl-D-sucrose, L-proline, 1-O-caffeoyl-(6-O-glucosyl)-β-D-glucose, xanthine, succinyladenosine, and xanthosine were upregulated in M3 and downregulated in M2. Notably, N-oleoylethanolamine was downregulated in both M1 and M3 but upregulated in M2.

### 2.4. Transcriptional Regulatory Network Analysis of Pollen Wall Development in Cabbage

In the model plant *Arabidopsis thaliana*, the DYT1-TDF1-AMS1-MS188-MS1 pathway regulates tapetum layer synthesis, with several genes in this network also involved in pollen wall formation and microspore development. The bHLH transcription factor ABORTED MICROSPORES (AMS), expressed in the *Arabidopsis* tapetum, is a key regulator of pollen wall development. AMS regulates several processes, including callose metabolism (CalS5), lipid metabolism (CYP703A2 and CYP704B1), sporopollenin formation (PKSB and TKPR1), tryphine production (EXLs and GRPs), and lipid transport (ABCG26) [29,30,31]. In the present study, we examined the expression of *Arabidopsis* homologous genes within this regulatory network in *Brassica* (Figure 4). Notably, the expression patterns of several genes in this pathway exhibited divergent trends between M2 and M3.

Callose synthases (CalSs) are critical for callose synthesis, and the expression of BoCalS5 was significantly reduced in both M2 and M3, leading to a substantial decrease in callose production. This reduction results in an abnormal pollen wall structure, consistent with the findings presented in Figure 1. The genes involved in phenolic metabolism, such as *BoPKSB* and *BoTKPR1*, as well as those related to lipid metabolism (*BoCYP704B1*, *BoCYP703A1*, and *BoCYP703A2*), precursor transport (*BoABCG26*), and sporopollenin biosynthesis (*BoTEK*, *BoAMS1*, *BoMS188*, and *Ms-cd1*), were downregulated in M1 and M2, but upregulated in M3. Based on our transcriptomic analysis, we hypothesize that the genes involved in lipid synthesis and sporopollenin biosynthesis are downregulated in the DGMS line and upregulated in the CMS line, thereby influencing pollen development through lipid regulation.

Notably, lipid transfer proteins (LTPs) are likely involved in transporting sporopollenin precursors. The genes *BoLTP11* and *BoLTP13* were upregulated in M1 and M2, but downregulated in M3. The upregulation of *BoLTP11* in the DGMS line can be inferred to downregulate lipid synthesis and metabolism-related genes. Furthermore, the formation of tryphine, which fills the cavities of the pollen exine, is influenced by extracellular lipase genes (EXLs) and glycine-rich proteins (GRPs). These factors may be regulated directly or indirectly by *MS188*. The expression of *BoMS188* was downregulated in both M1 and M2, and further downregulated in M3, likely contributing to the accumulation of *BoEXLs* and *BoGRPs* in M2, while their expression was downregulated in M3. This regulatory dynamic influences pollen wall development in both DGMS and CMS lines.

### 2.5. Carbohydrate Metabolic Pathway Transcriptome Metabolome Integration Analysis

To identify the key regulatory pathways potentially involved in both the DGMS and CMS systems of *Brassica oleracea*, we comprehensively analyzed the transcriptomic and metabolomic data. Based on this analysis, we performed a KEGG pathway enrichment analysis of the top 30 most significantly enriched pathways among the DEGs and DAMs identified in this study (Appendix A). The integrated transcriptomic and metabolomic analysis revealed that the primary metabolic pathways enriched in both the CMS and DGMS systems of *Brassica oleracea* were those related to carbohydrate metabolism, including glycolysis/gluconeogenesis (ko00010), carbon metabolism (ko01200), starch and sucrose metabolism (ko00500), and the TCA cycle (ko00020).

Metabolites are the ultimate products of gene expression regulation, and their analysis provides crucial insights into the metabolic pathways underlying cellular processes such as CMS. In this study, we integrated data from glycolysis/gluconeogenesis, carbon metabolism, starch and sucrose metabolism, and the TCA cycle to elucidate the role of carbohydrate metabolism in CMS formation (Figure 5). By combining transcriptomic and metabolomic datasets, we were able to identify 105 DEGs and 10 DAMs involved in carbohydrate metabolism pathways.

In both DGMS and CMS lines, key regulatory enzymes such as HXK1, PCK2, and IDH exhibited downregulated expression. HXK1, the initial enzyme in glycolysis, catalyzes the phosphorylation of glucose into glucose-6-phosphate (G6P), a critical step in energy production. Its downregulation impedes glycolysis, disrupting the energy supply required for proper pollen development. In DGMS line, the reduced expression of HXK1 diminishes glycolytic flux, while PCK2, a key enzyme in gluconeogenesis, contributes to increased pyruvate synthesis, affecting the overall energy balance. Additionally, IDH downregulation alters the flow of intermediates into the TCA cycle, further impacting cellular metabolism.

In CMS line, glycolytic enzymes such as PFK3, PFK2, PFK7, and L-aspartate oxidase were upregulated in the M3 samples. This regulation impaired the production of glucose-6-phosphate and fructose-6-phosphate, decreasing the accumulation of these intermediates and elevating glucose levels, as confirmed through GC-MS analysis. The increased expression of fructose–bisphosphate aldolase 6 (*Bol037713*), coupled with the reduced expression of phosphoglycerate mutase-like protein (Bol005481), suggests the enhanced accumulation of glyceraldehyde-3-phosphate in CMS line. Furthermore, the downregulation of PKp3 and PKP-BETA1 pyruvate kinase in the gluconeogenesis pathway reduced acetyl-CoA production, limiting its entry into the TCA cycle. In the TCA cycle, CMS line exhibited decreased IDH (Bol029959) expression and suppressed oxaloacetate breakdown due to PCK2 downregulation, increasing the accumulation of intermediates such as 2-oxoglutarate, succinyl-CoA, succinate, fumarate, and malate. The metabolomic analysis confirmed a significant increase in fumarate content in the CMS line compared with the DGMS line, which accumulated comparatively less.

### 2.6. The Critical Role of Lipid Biosynthesis and Metabolism in Male Gametophyte Development

Lipid biosynthesis and metabolism play a crucial role in synthesizing the pollen wall, an essential process during male gametophyte development [32]. To investigate the involvement of lipid metabolic pathways in regulating male sterility, we constructed a detailed pathway map illustrating the key genes and differentially accumulated metabolites associated with fatty acid metabolism (Figure 6). Within plastids, fatty acids are synthesized by two enzyme systems: acetyl-CoA carboxylase and fatty acid synthase [33]. These enzymes catalyze the initial and elongation steps of fatty acid synthesis, respectively. Our analysis revealed that most differentially accumulated metabolites involved in fatty acid metabolism showed significant reductions during both the M2 and M3 stages. Notably, this reduction was more pronounced in the M3 stage, suggesting a stage-specific exacerbation of lipid metabolic disruptions.

In the DGMS line, *Bol039786* downregulation reduces the levels of hexadecanoyl-CoA and palmitic acid, increasing the accumulation of acyl-CoA within the mitochondria. Additionally, *Bol040329* downregulation disrupts the synthesis of long-chain fatty acids, which causes insufficient lipid biosynthesis and impairs the normal development of the pollen wall.

In the CMS line, the concentrations of several differential metabolites, including decanoic acid, octanoic acid, myristic acid, palmitic acid, and palmitoleic acid, are notably reduced. Furthermore, the FabG gene, which is involved in fatty acid biosynthesis, is upregulated (*Bol040329*), suggesting that increased FabG activity may deplete important cofactors, such as NADPH, disrupting other essential metabolic pathways. The overactivation of this enzyme could lead to an imbalance in the pollen wall matrix, ultimately affecting fertility regulation in the CMS line. In the mitochondrial matrix, the reduced levels of palmitic acid and hexadecanoyl-ACP may impair the efficient utilization of acetyl-CoA, resulting in its accumulation in the intermembrane space and exacerbating the mitochondrial metabolic imbalance. Moreover, this imbalance likely induces feedback inhibition in the citric acid cycle, further amplifying the metabolic dysregulation.

These findings underscore the critical role of fatty acid metabolism in regulating cytoplasmic male sterility, emphasizing how disruptions in lipid biosynthesis pathways contribute to pollen wall abnormalities and male sterility phenotypes.

### 2.7. Disruption of Phenylpropanoid Metabolism and Synthesis (Flavonoids) in CMS Lineage Development

KEGG enrichment analysis of the DEGs and DAMs revealed that the most significantly affected metabolic pathways are those associated with secondary metabolism. In particular, the phenylpropanoid biosynthesis pathway was notably influenced by the DEGs, suggesting that regulating phenylpropanoid synthesis and degradation is a key factor driving the formation of male sterility. A total of 420 DEGs from the three cabbage materials were mapped to the phenylpropanoid biosynthesis pathway. By integrating metabolic network analysis and gene expression data, we were able to construct a detailed phenylpropanoid metabolism pathway (Figure 7), underscoring its pivotal role in regulating male sterility.

In the DGMS metabolic pathway, the early-stage accumulation of phenylpyruvic acid is reduced, and L-phenylalanine metabolites are diminished. The upregulation of C4H (*Bol004608* and *Bol004610*) and CHS (*Bol008652*), the key enzyme in flavonoid biosynthesis, contributes to the accumulation of hesperetin, which, in turn, enhances the plant’s antioxidant capacity.

In the CMS metabolic pathway, the reduction in phenylpyruvic acid accumulation is even more pronounced compared with the DGMS line. However, the increased expression of *Bol032870* and *Bol035895* leads to the accumulation of L-phenylalanine metabolites in the DGMS line. Concurrently, the downregulation of PAL and 4CL, which are involved in cellulose biosynthesis, decreases p-coumaroyl-CoA. Both p-coumaroyl-CoA and ferulic-CoA are critical for flavonoid formation, thus affecting sporopollenin biosynthesis and suggesting that inhibited cellulose synthesis may be a significant factor in abnormal pollen wall structures. Furthermore, the accumulation of fumaric acid may exacerbate the sterility phenotype in CMS. In downstream pathways, the upregulation of Type III polyketide synthase A (*Bol025267*) in M3 accelerates the consumption of phoretin, while phoretin significantly accumulates in M2, resulting in metabolic differences in phlorizin between M2 and M3. These findings highlight key regulatory points in the phenylpropanoid pathway that influence male sterility in both DGMS and CMS systems.

### 2.8. The Unique Role of Vitamin B6 Metabolism in the CMS Lineage

Vitamin B6, an essential cofactor involved in numerous enzymatic reactions, plays a unique role in the CMS system. Its involvement in amino acid metabolism, nucleic acid synthesis, and stress response mechanisms is well documented. However, in CMS materials, vitamin B6 appears to be closely linked to the regulation of energy metabolism and mitochondrial function, both which are crucial for proper pollen development. Figure 8 illustrates the metabolic pathways associated with vitamin B6, particularly its various forms, such as pyridoxal, pyridoxine, and their phosphorylated derivatives, including pyridoxal 5′-phosphate.

In CMS line, the expression of the pyridoxal 5′-phosphate synthase gene *Bol008957* was significantly upregulated, leading to an increase in pyridoxal 5′-phosphate accumulation. In contrast, the downregulation of cholinesterase-related genes (*Bol009758*, *Bol013254*) resulted in a significant reduction in the accumulation of metabolites such as 4-pyridoxate, pyridoxal, and pyridoxine. This alteration in the metabolic profile suggests a disruption in the synthesis and regulation of vitamin B6 forms, which could impact critical pathways involved in pollen development. The accumulation of pyridoxal 5′-phosphate, catalyzed by the upregulation of pyridoxal 5′-phosphate synthase, may enhance cofactor availability for enzymatic reactions that support energy metabolism. However, the downregulation of cholinesterase-related genes and the corresponding reduction in 4-pyridoxate, pyridoxal, and pyridoxine levels suggests that a metabolic imbalance is induced, potentially hindering normal metabolic fluxes necessary for pollen formation. This imbalance may further lead to distinct manifestations observed in CMS and DGMS lines, in addition to the male sterility phenotype, such as an increased number of dead buds, thereby highlighting the intricate relationship between vitamin B6 metabolism and mitochondrial function during male gametophyte development.

### 2.9. qRT-PCR Analysis

To further validate the DEGs identified in the transcriptome analysis, quantitative real-time PCR (qRT-PCR) experiments were conducted. Key genes associated with male sterility, including those involved in pollen development (*Bol033811 and Bol01793*), sugar metabolism (*Bol037713*), lipid synthesis (*Bol040329* and *Bol039786*), phenylpropanoid metabolism (*Bol004610*, *Bol004608*, and *Bol025267*), and vitamin B6 metabolism (*Bol008957* and *Bol013254*), were selected for analysis. The qRT-PCR experiments utilized the same RNA samples as the transcriptome data, and primers consistent with those used in the transcriptome analysis were employed to ensure data consistency and accuracy. The results showed that the majority of the qRT-PCR data aligned with the transcriptome findings, further confirming the significant role of DEGs in male sterility (Figure 9).

## 3. Discussion

We conducted a comparative analysis of metabolomics and transcriptomics between two male-sterile lines, DGMS and CMS, and their corresponding maintainer line. Electron microscopy observations revealed that DGMS exhibited earlier male sterility than CMS. During the tetrad stage, microspore outer wall degradation was absent in DGMS, leading to abnormal microspore development. In contrast, CMS microspore development remained largely normal up to the uninucleate pollen stage. However, after the uninucleate stage, the degradation and vacuolization of the callose layer became evident, along with the downregulation of callose-layer-related genes (Figure 1B and Figure 4). We analyzed the expression of the genes associated with pollen wall development in both sterile lines and identified significant differences. The number of DEGs and differentially accumulated metabolites (DAMs) in the M1_vs_M3 comparison was 2–3 times higher than in M1_vs_M2, particularly in the CMS line. This suggests that more genes and metabolites are involved in transcriptional regulation and secondary metabolic pathways in the CMS line. In terms of carbohydrate metabolism, fatty acid metabolic pathways, phenylpropanoid biosynthesis, and vitamin B6 metabolism, the CMS line exhibited a more complex and extensive range of energy and substance metabolism than the DGMS line. This imbalance may lead to the distinct manifestations observed in CMS and DGMS line in addition to the male sterility phenotype, such as an increased number of dead buds, highlighting the intricate relationship between vitamin B6 metabolism and mitochondrial function during male gametophyte development. In conclusion, this study provides an initial exploration of the molecular mechanisms differentiating the DGMS and CMS lines through the application of omics technologies, establishing a foundation for the development of future hybrid breeding strategies in cabbage.

### 3.1. Differences in the Regulatory Network of Pollen Wall Development Between the DGMS and CMS Lines Contribute to the Differences in Their Respective Sterility Phenotypes

The pollen wall is a dense structure located on the surface of plant gametophytes, playing a critical role in maintaining microspore stability during pollen development, protecting against various biotic and abiotic stresses, and facilitating recognition by the stigma [29]. Pollen wall development is a highly coordinated process influenced by various genetic and metabolic pathways. During the pollen maturation stage, although different plant species exhibit distinct pollen surface morphologies, mature pollen grains generally consist of three layers: the outer exine, the inner intine, and the tryphine [34,35]. The outer wall is composed of an outer and inner layer, with sporopollenin being the primary constituent. The callose layer is the main structure responsible for synthesizing sporopollenin precursors. Previous studies on *Arabidopsis* have identified a genetic regulatory pathway involving DYT1-TDF1-AMS-MS188-MS1, which regulates the development of the callose layer and pollen wall. Transcription factors AMS and MS188 are the key regulators of sporopollenin biosynthesis gene expression [36]. Genes associated with AMS transcription, such as *BoPKSB*, *BoTKPR1*, *BOCYP704B1*, *BoCYP703A1*, *BoCYP703A2*, *BoABCG26*, *BoEXLs*, *BoGRPs*, *BoTEK*, and *BoAMS1*, are downregulated in the DGMS line and upregulated in the CMS line. These differential expression patterns lead to defects in sporopollenin synthesis in the DGMS line. The ABC transporter and LTPs are likely implicated in the transport of sporopollenin precursors. Notably, BoLTP11 upregulation in the DGMS line facilitates the transport of sporopollenin precursors, which aligns with previous findings and the phenotypic observation in Figure 1, where microspore development in DGMS tetrads halts at the tetrad stage, preventing the formation of a normal pollen outer wall.

*BoEXLs* and *BoGRPs* exhibit contrasting expression levels in the DGMS and CMS lines. These genes not only regulate the synthesis of glycine-rich proteins (GRPs), but also coordinate the biosynthesis of sporopollenin precursors and extracellular lipases (EXLs), both of which are crucial for the stable formation of the pollen wall. In the DGMS line (M2), the upregulation of *BoEXLs* and *BoGRPs* supports the hypothesis proposed by Han et al. [37] that MS-cd1 may function as an activator of pollen exine development. Conversely, in the CMS line (M3), *BoEXLs* and *BoGRPs* are significantly downregulated, suggesting disruptions in the biosynthesis and transport of critical lipid and glycoprotein precursors required for exine development. This finding aligns with the observed phenotype of CMS20-2-5, which lacks a normal pollen exine (Figure 1B). These differential expression patterns highlight substantial mechanistic differences between CMS and DGMS sterility. The CMS line appears to suffer from broader mitochondrial and metabolic pathway disruptions, which impair the coordinated expression of exine-associated genes.

In terms of the developmental stages of flower formation, DGMS line exhibit earlier and more complete sterility than CMS line, with the expression of relevant genes potentially becoming aberrant as early as before tetrad formation. In contrast, CMS line show little difference from their maintainer line before the unicellular stage, with pollen gradually collapsing and degrading after this stage. These differences in developmental timing lead to distinct regulatory mechanisms, allowing us to conclude that the shared DEGs and metabolites between the DGMS and CMS lines likely determine phenotypic traits after the unicellular stage. The DEGs unique to the DGMS line likely regulate fertility-related genes before tetrad formation, while the DEGs unique to CMS line may govern broader regulatory pathways beyond male sterility. This novel finding provides valuable insights into the development of male-sterility-based breeding strategies.

### 3.2. Carbohydrate Metabolism and Fertility

Carbohydrate metabolic pathways are essential for energy production and developmental processes, making them a focal point of this study. Through the integration of transcriptomic and metabolomic analyses, we identified significant alterations in carbohydrate metabolic pathways, including glycolysis, the tricarboxylic acid (TCA) cycle, and starch metabolism. Several genes associated with glycolysis and the TCA cycle, such as HXK1, PFK3, and GAPC1, exhibited distinct expression profiles between the CMS and DGMS lines. Notably, the expression levels of HXK1, PCK2, and IDH were significantly reduced in both the CMS and DGMS systems. These enzymes are the key regulators of carbohydrate metabolism, and their downregulation suggests potential disruptions in these critical metabolic processes.

HXK1 catalyzes the first step of glycolysis, converting glucose into G6P. It has been well established that HXK1 plays a crucial role in pollen development and male sterility. Its function in glucose metabolism is intricately linked to the energy demands of gametophyte development and pollen wall formation. PCK2 catalyzes the conversion of oxaloacetate into phosphoenolpyruvate (PEP), a critical step in gluconeogenesis that enables the synthesis of glucose from intermediates such as lactate, pyruvate, and amino acids; this enzyme is essential for maintaining energy homeostasis. Similarly, IDH directly regulates the concentration of 2-oxoglutarate, a key intermediate that controls the entry of metabolites into the TCA cycle. In summary, the downregulation of HXK1, PCK2, and IDH suggests disruptions in energy metabolic pathways in these male-sterile lines, potentially impairing the energy supply and metabolic balance necessary for proper pollen development.

In the DGMS line, carbohydrate metabolism appears to be less disrupted than in the CMS line. Key genes, such as PFK3 and PKP-BETA1 (pyruvate kinase), are upregulated, ensuring a steady flux through glycolysis and downstream metabolic pathways. This relative metabolic stability aligns with the DGMS system’s reliance on nuclear gene regulation rather than mitochondrial dysfunction, highlighting the distinct molecular mechanisms underlying sterility in these two systems.

In the CMS line, key genes associated with glycolysis, such as HXK1 and PFK7, are predominantly downregulated, disrupting glycolysis and the subsequent production of acetyl-CoA, a precursor for lipid biosynthesis. This results in reduced levels of critical intermediates, including PEP and acetyl-CoA. These metabolic bottlenecks may impair energy balance, leading to defects in pollen wall development and male sterility, which is consistent with the observed increase in fumarate accumulation and decreased ATP production. These metabolic alterations suggest systemic mitochondrial dysfunction, supporting the established role of mitochondria in the CMS phenotype. Both the CMS and DGMS lines exhibit changes in the pentose phosphate pathway and sucrose metabolism, which are essential for providing precursors for cell wall biosynthesis and pollen development. However, these changes are more pronounced in the CMS line, indicating a broader scope of metabolic disruption. Overall, the integration of transcriptomic and metabolomic data reveals significant alterations in carbohydrate metabolism in both the CMS and DGMS lines, resulting in substantial changes in energy production and metabolite accumulation. These metabolic disturbances are likely central to the observed male sterility phenotype, underscoring the critical role of carbohydrate metabolism in regulating pollen development and fertility.

### 3.3. Impaired Lipid Metabolic Pathways Are Closely Related to Pollen Wall Mechanisms

During gametophyte development, lipid synthesis and metabolism play crucial roles in pollen wall formation [32]. Lipids and their derivatives, such as fatty acids, waxes, and phospholipids, are integral components of the pollen wall. The biosynthesis of fatty acids is initiated by acetyl-CoA carboxylase, an enzyme that facilitates the conversion of acetyl-CoA into malonyl-CoA via a carboxylation reaction, a process that is ATP-dependent [38]. Subsequently, some of these fatty acids are further transported and elongated in the endoplasmic reticulum. Through pathway enrichment analysis, Zhang et al. [39] demonstrated that proteins associated with defective pollen wall formation are predominantly involved in the de novo synthesis and metabolism of fatty acids. As depicted in the fatty acid metabolism pathway (Figure 6), fatty acids are esterified into acyl carrier proteins (ACPs), the predominant form within plastids. These long-chain acyl-ACPs are then cleaved by acyl-ACP thioesterases and transported from plastids to the endoplasmic reticulum, where they are further extended onto CoA. The biosynthesis of long-chain fatty acids begins with acetyl-CoA, which is converted into malonyl-CoA by acetyl-CoA carboxylase and subsequently elongated into long-chain acyl-ACP. These long-chain acyl-ACPs are essential precursors for the production of lipids, waxes, and monomeric proteins that are critical components of the pollen wall.

Significant differences were observed between the M2 (DGMS20-2-5) and M3 (CMS20-2-5) lines. The M3 line exhibited higher levels of certain fatty acids yet displayed lower ATP levels, indicative of mitochondrial dysfunction. These findings suggest that the fatty acid metabolism pathway plays a critical role in the M3 (CMS20-2-5 line). Moreover, in the CMS line (M3), there was a notable disruption in the biosynthesis and elongation of fatty acids, reflected in the downregulation of key genes such as *Bo004398* and *Bo021081*, alongside a significant reduction in intermediate metabolites, including palmitic acid. Myristic acid, which contributes to the synthesis of long-chain fatty acids, is an essential component of the pollen wall (exine) [40]. Myristic acid deficiency in M3 leads to abnormal pollen wall development and male sterility. Studies have shown that myristic acid is involved in the biosynthesis of waxes in the pollen exine, as well as in lipid transport and deposition during anther development. In the CMS line, disruptions in fatty acid metabolism (including the synthesis and elongation of myristic acid) likely compromise the pollen wall structure and reduce the lipid content within the anther tissue. Impaired mitochondrial function or lipid transport mechanisms ultimately lead to male sterility. These findings underscore the pivotal role of fatty acid metabolism in pollen wall biosynthesis and suggest that impaired lipid metabolic pathways are closely linked to the mechanisms underlying male sterility.

### 3.4. Importance of Phenylpropanoid Compounds in Maintaining Fertility

Phenylpropanoids are a class of naturally occurring compounds characterized by a phenyl ring attached to a three-carbon side chain (C6-C3). Typically possessing a phenolic structure, these compounds are considered phenolic substances and represent the largest group of secondary metabolites produced by plants. Phenylpropanoids primarily function in response to biotic and abiotic stresses, including injury and environmental factors. The molecular basis for their protective role in plants lies in their antioxidant properties and the ability to scavenge reactive species. These phenolic compounds are among the key bioactive constituents in plants. As illustrated in the phenylpropanoid metabolic pathway (Figure 7), PAL is the first crucial enzyme in this pathway, catalyzing the conversion of L-phenylalanine into cinnamic acid. The activity and gene expression of PAL directly influence the synthesis of downstream products, including flavonoids, cellulose, and antioxidant phenolic compounds, underscoring its pivotal role in phenylpropanoid biosynthesis.

In the DGMS line, most of the DEGs align with those in the maintainer line, showing a relatively moderate decline in energy metabolism. This suggests that male sterility in DGMS may involve more localized or specific regulatory mechanisms. Both the DGMS and CMS lines exhibit disruptions in the phenylpropanoid metabolic pathway, particularly in the biosynthesis of secondary metabolites such as flavonoids and lignins. These disruptions likely impair pollen development and contribute to male sterility.

In the CMS line, the expression of PAL genes is significantly reduced, leading to the overall suppression of phenylpropanoid synthesis. Fumaric acid levels are elevated, while ATP levels are diminished—compounds essential for pollen wall formation and the mitigation of oxidative stress—indicating mitochondrial dysfunction and a widespread energy deficiency, which is associated with reduced ATP synthesis in dysfunctional mitochondria. These findings align with transcriptomic data showing the downregulation of mitochondrial genes in the CMS line. Secondary metabolites, particularly flavonoids and phenylpropanoids, are notably reduced, compounds that are essential in pollen wall formation and mitigating oxidative stress. Furthermore, genes encoding the enzymes involved in flavonoid biosynthesis, such as CHS, are also significantly downregulated, correlating with a reduction in flavonoid levels. These results underscore the critical role of phenylpropanoids in maintaining fertility, suggesting that the targeted manipulation of these genes and associated pathways could provide strategies for restoring fertility or enhancing sterility in hybrid seed production. Collectively, these data highlight severe disruptions in phenylpropanoid metabolism in the CMS line, with a significant upregulation of flavonoid biosynthesis genes and the suppression of lignin-related pathways. These findings provide valuable insights into the key metabolic adjustments underlying male sterility mechanisms, offering biochemical foundations for understanding sterility.

### 3.5. Future Research Directions and Applications

The findings of this study hold significant implications for both fundamental research and practical applications. Future investigations should focus on functionally validating the identified candidate genes and metabolites to confirm their roles in male sterility. Advanced technologies, such as CRISPR/Cas9 gene editing and metabolite supplementation experiments, offer promising approaches to gaining deeper insights into the underlying mechanisms.

From an applied perspective, the differential genes identified in this study have potential utility in breeding programs to facilitate the efficient selection of sterile or fertile lines. Furthermore, the knowledge gained from this research could inform the development of novel hybrid seed production strategies. For instance, synthetic male sterility systems could be designed, or existing CMS and DGMS systems could be improved to enhance their efficiency and adaptability.

## 4. Materials and Methods

### 4.1. Plant Materials and Sample Collection

Cabbage CMS line CMS20-2-5 (M3) and DGMS20-2-5 (M2) and the latter’s maintainer line 20-2-5 (M1) were utilized in the present study, with the maintainer line serving as the control. The plants were initially grown at the experimental base of the Institute of Vegetables and Flowers, Chinese Academy of Agricultural Sciences, Beijing, in the autumn of 2023. In winter, they were transferred to a greenhouse, and samples were collected following the initiation of bud formation in April 2024. Incompletely opened flower buds from the CMS and DGMS lines were collected, immediately placed on ice, and rapidly frozen using liquid nitrogen before being stored at −80 °C for further analysis. All samples were collected at 11:00 AM.

### 4.2. Cytological Microscopy

Cytological analyses of anthers from CMS and DGMS plants at both the tetrad and mononuclear pollen grain stages were conducted to investigate the differences in male gametophyte development between these two plant types. For transmission electron microscopy (TEM), floral buds from cabbage at the tetrad stage were harvested, rinsed with 0.1 M phosphate buffer, and then embedded in epoxy resin. Ultrathin sections (50-70 nm) were prepared using a Leica EM UC7 ultramicrotome (Leica Microsystems, Brussels, Belgium), and stained with uranyl acetate and lead citrate. The sections were then examined under a HITACHI Regulus 8100 field-emission scanning electron microscope (Hitachi, Tokyo, Japan) (Figure 1).

### 4.3. RNA Extraction and Transcriptome Sequencing

Total RNA was extracted from various floral tissue components using the Plant RNA Kit (Tiangen, Beijing, China). To ensure the RNA samples met the necessary standards for further analysis, their quality was thoroughly evaluated. The concentration, purity, and integrity of the total RNA were assessed using a spectrophotometer (NanoDrop 2000, Thermo Fisher Scientific, Wilmington, DE, USA) and an Agilent Bioanalyzer 2100 system (Agilent Technologies, Santa Clara, CA, USA). After RNA extraction, poly(A) mRNA was isolated using oligo(dT)-coated beads. The mRNA was then fragmented into short segments with the addition of a fragmentation buffer. For cDNA synthesis, the first strand was generated using random hexamers as primers, followed by second-strand cDNA synthesis with the addition of a buffer, dNTPs, RNase H, and DNA polymerase I. The resulting cDNA was purified using AMPure XP beads (Beckman Coulter, Brea, CA, USA) and processed according to the Illumina library preparation protocol. The cDNA underwent end repair, phosphorylation, and the addition of an ”A” base, in line with the Illumina protocol for library construction. The final library had an insert size of 200–300 bp. cDNA fragments were selected using 2% Low-Range Ultra agarose, and amplified via PCR for 15 cycles with Phusion DNA polymerase (New England Biolabs, Boston, MA, USA). After quantification with the TBS380 system (Turner BioSystems, San Jose, CA, USA), the RNA sequencing libraries were subjected to single-end sequencing on a NovaSeq 6000 sequencer (Illumina, San Diego, CA, USA), generating 2 × 150 bp paired-end reads. Three biological replicates were prepared for each sample, with RNA extraction and sequencing performed by Beijing BioMarker Biotechnology Co., Ltd. (Beijing, China). The RNA-Seq data analyzed in this study were deposited in the NCBI SRA database (BioProject: PRJNA1190600).

### 4.4. Metabolite Extraction and Analysis Methods

The anthers were subjected to vacuum freeze-drying using a lyophilizer (Scientz-100F lyophilizer, Scientz, Ningbo, China) and subsequently ground into a fine powder (30 Hz for 1.5 min) with a mill (MM 400, Retsch, Haan, Germany). A 100 mg sample of the powder was weighed and dissolved in 1.2 mL of 70% methanol. The solution was vortexed for 30 s every 30 min, for a total of six cycles. The dissolved sample was then stored overnight at 4 °C. After centrifugation (10,000× *g* for 10 min), a 200 μL aliquot of the supernatant was carefully transferred into a 2 mL injection vial. For quality control (QC) purposes, 20 μL from each sample was pooled to prepare a QC sample. The remaining 200 μL of the supernatant was used for UPLC-MS/MS analysis. The metabolomics analysis was performed using a Waters Acquity I-Class PLUS ultra-performance liquid chromatography (UPLC) system (Waters Corporation, Milford, MA, USA) coupled with a Waters Xevo G2-XS QTof high-resolution mass spectrometer. Chromatographic separation was achieved using a Waters Acquity UPLC HSS T3 column (1.8 μm, 2.1 × 100 mm), and data were acquired using MassLynx V4.2 (Waters Corporation, Milford, MA, USA). Peak extraction and alignment were performed using Progenesis QI (Waters Corporation, Milford, MA, USA), and metabolites were identified using the METLIN database and a self-built library from Biomarker Technologies, with theoretical fragment identification. The mass deviation of the identified ions was within 100 ppm. The assay was performed by Beijing BioMarker Biotechnology Co., Ltd.

### 4.5. Bioinformatic and Statistical Analysis

To accurately reflect the transcript expression levels, it is essential to normalize the number of mapped reads and the transcript length. FPKM (Fragments Per Kilobase of transcript per Million Fragments Mapped) [41] was used as a measure of transcript or gene expression, calculated using the maximum flow algorithm in StringTie (Johns Hopkins University, Baltimore, MD, USA). The FeatureCounts software (FeatureCounts v1.5.0-P3, WEHI, Melbourne, Australia) computes the number of reads mapped to each gene and subsequently calculates the FPKM for each gene based on its length. Differentially expressed genes at the bud stage in various cabbage materials were identified using the DESeq2 R package [42]. The FC represents the ratio of the expression between two groups of samples, while FDR is calculated by adjusting for the *p*-values of differential significance. The FDR was determined by correcting *p*-values to assess the significance of the differences. A fold change of ≥2 and an FDR of <0.01 were used as criteria to identify significant differential expression.

The raw sequencing reads were processed further using a bioinformatics pipeline provided by the BMK-Cloud online platform (www.biocloud.net, accessed on 7 September 2024, BioMarker Technologies, Beijing, China). Hierarchical clustering heatmaps, based on the Euclidean distance algorithm, were generated using the Heatmap package in TBtools [43]. For functional annotation, GO and KEGG pathway enrichment analyses were performed using the Database for Annotation, Visualization, and Integrated Discovery (DAVID, version 6.8) [44]. GO terms, including biological processes, molecular functions, and cellular components, as well as KEGG pathways with *p*-values less than 0.05, were considered significantly enriched. PCA was conducted using SIMCA-P (version 14.1, Umetrics, Umea, Sweden) and visualized with the R package ggplot2. For correlation analysis, the Pearson correlation coefficient was computed using R functions, with log2-transformed ion intensity values as input data.

### 4.6. Real-Time Quantitative PCR Analysis

RNA was extracted using the TIANGEN RNA extraction kit, following the procedure outlined in Section 4.3. Reverse transcription was performed using the TIANGEN FastQuant RT Kit (Tiangen, Beijing, China). The qRT-PCR procedure, based on Han et al. [37], was carried out on a CFX96 Touch™ Real-Time PCR System (Bio-Rad, Hercules, CA, USA), with slight modifications: primers were designed using Primer 5.0, and a 20 μL RT-qPCR reaction mixture was prepared with 10 μL SYBR Green PCR Master Mix (Vazyme Biotech Co., Ltd., Nanjing, China), 1 μL cDNA, 0.5 μL each of forward and reverse primers, and ddH_2_O to a final volume of 20 μL. Three technical replicates were performed for each sample. The expression levels of DEGs were calculated using the 2^−∆∆Ct^ method, with *Actin* serving as the internal reference gene. Primers were designed using PRIMER 5 (Appendix A). Data analysis and plotting were conducted using GraphPad Prism 8.4 (GraphPad Software, San Diego, CA, USA).

## 5. Conclusions

This study demonstrates that integrating transcriptomics and metabolomics provides a comprehensive framework for unraveling the regulatory networks underlying male sterility in cabbage. Five key regulatory pathways—pollen wall development, carbohydrate metabolism, lipid biosynthesis, phenylpropanoid metabolism, and vitamin B6 metabolism—are pivotal in male sterility. Compared with the DGMS line, the CMS line exhibits more differentially enriched genes and metabolites, indicating more extensive metabolic disruption. Genes uniquely expressed in the DGMS line, compared with the CMS line, govern male sterility at early developmental stages. In contrast, the CMS line shares differentially expressed genes with the DGMS line that control the pollen abortion pathway, which occurs after monokaryotic pollen grains form. Additionally, the genes differentially expressed in the CMS line, but not in the DGMS line, contribute to late-stage sterility and bud abscission. The overlapping regulatory networks emphasize the common key points of sterility, while the distinct pathways underscore the regulatory mechanisms specific to each sterility type. These insights lay the groundwork for further functional studies and breeding strategies aimed at targeting male sterility in cruciferous crops.

## Figures and Tables

**Figure 1 ijms-26-01259-f001:**
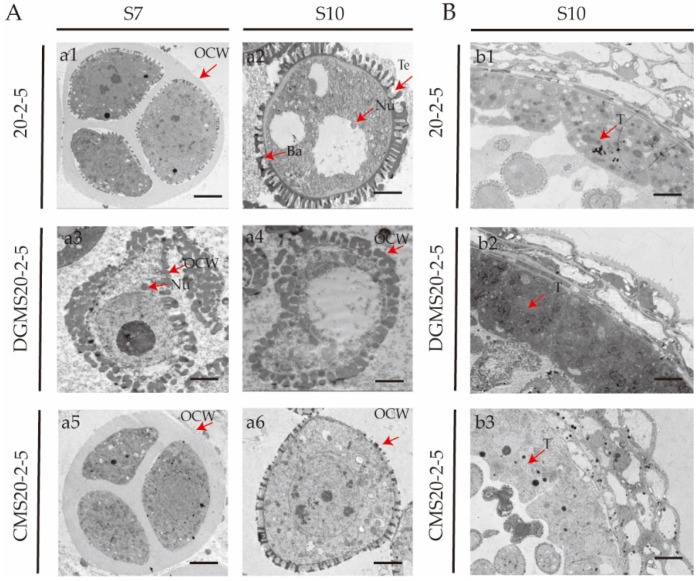
Transmission electron microscopy (TEM) analysis of microspores from 20-2-5, DGMS20-2-5, and CMS20-2-5 at anther stages S7 and S10. (**A**) Comparison of pollen grains. (**B**) Comparison of the exine structure of pollen. T, tapetum; OCW, outer cell wall; Te, tectum; Ba, bacula; Nu, nucleus; scale bar = 2 μm.

**Figure 2 ijms-26-01259-f002:**
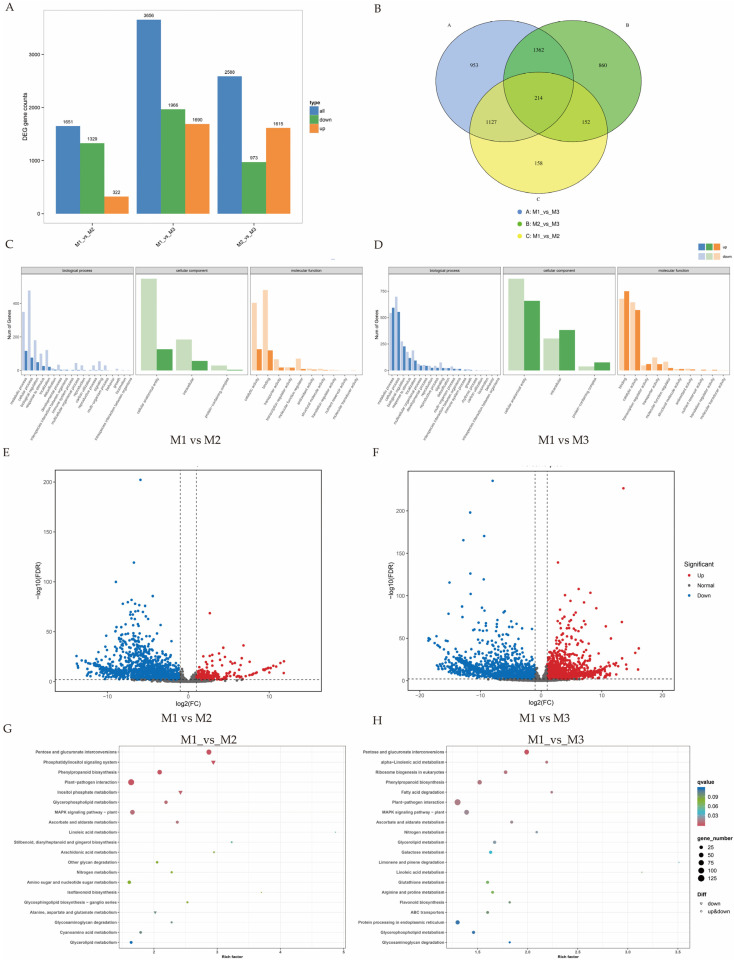
Distribution characteristics of differentially expressed genes (DEGs) in M1, M2, and M3. (**A**) Number of DEGs between comparison pairs. (**B**) Venn diagram illustrating the distribution of differentially expressed genes in the three groups. (**C**) GO annotation classification of differentially expressed genes in M1_vs_M2. (**D**) GO annotation classification of differentially expressed genes in M1_vs_M3. (**E**,**F**) Volcano plot of differential gene expression. (**G**,**H**) KEGG enrichment bubble plot of differentially expressed genes, where each circle represents a KEGG pathway. The vertical axis indicates the pathway name, and the horizontal axis represents the Rich factor, which is the ratio of the proportion of genes in the differentially expressed genes annotated to a pathway to the proportion of all genes annotated to that pathway.

**Figure 3 ijms-26-01259-f003:**
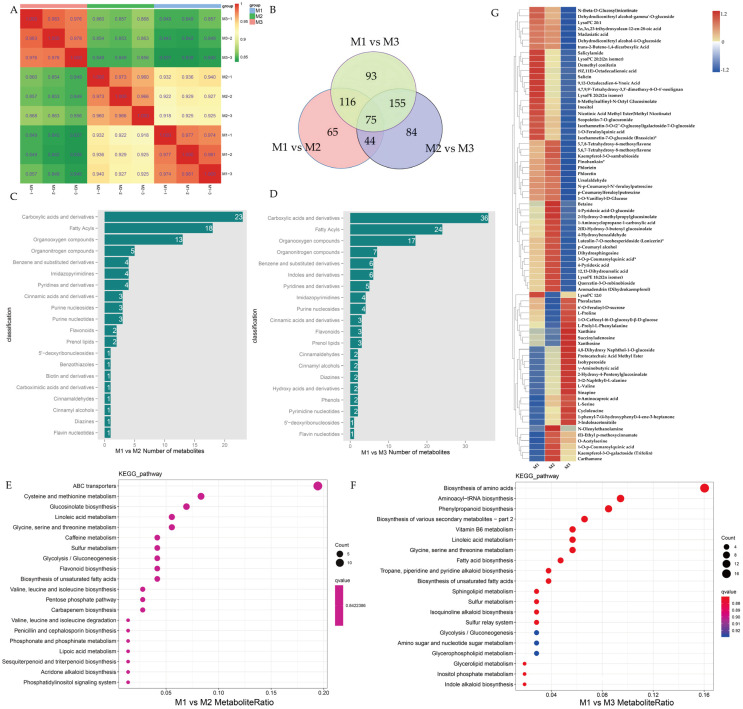
(**A**) Inter-sample correlation analysis. (**B**) Venn diagram illustrating the distribution of differential metabolites across the three groups. (**C**,**D**) Statistical analysis of differential metabolites based on annotations from the HMDB database. (**E**,**F**) KEGG enrichment analysis of differential metabolites. (**G**) Clustering heatmap of the 75 shared DAMs across M1, M2, and M3.

**Figure 4 ijms-26-01259-f004:**
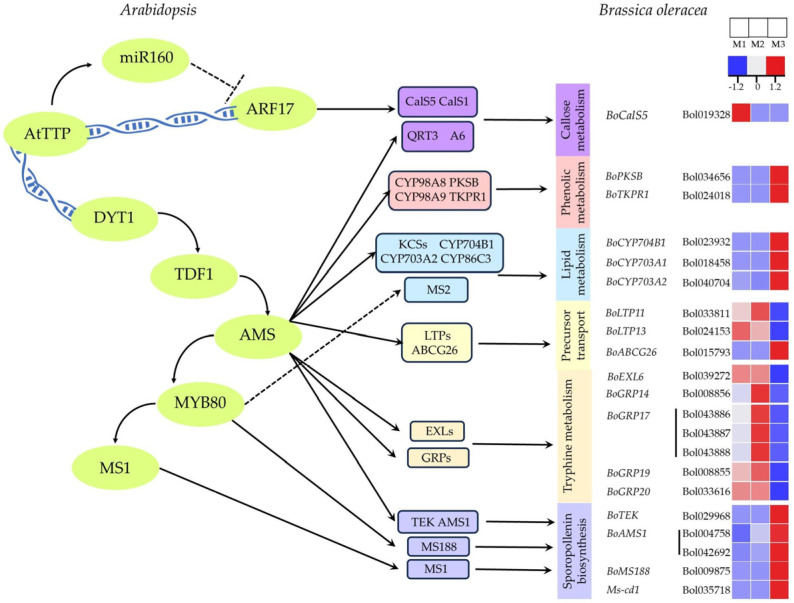
Transcriptional regulatory network of the genes involved in pollen wall development in *Brassica oleracea* based on the Arabidopsis thaliana model. Solid arrows indicate direct positive regulation supported by promoter binding, while dotted arrows represent positive regulation inferred from transcriptomic data. Positive and negative regulatory interactions are depicted by arrows and lines with bars, respectively. CalS5: callose synthase 5; CalS1: callose synthase 1; AtTTP: CCCH-type zinc finger protein; miR160: microRNA160; *DYT1*: *dysfunctional tapetum 1*; *TDF1*: *defective in tapetal development and function 1*; ARF17: auxin response factor 17; *AMS*: *aborted microspores*; MYB80: MYB transcription factor 80; *MS1*: *male sterility 1*; *MS2: male sterility 2*; *TEK*: *transposable element silencing* via *AT-hook*; AtTTP: CCCH zinc finger protein; LTPs: lipid transfer proteins; ABCG26: ATP binding cassette transporter; PKSB: polyketide synthases; EXLs: extracellular lipase; QRTA: Quantitative Resistance to Alternaria; KCs: Potassium Channels; LTPs: Lipid Transfer Proteins; ABC26: ATP-Binding Cassette Transporter 26; GRPs: Glycine-Rich Proteins.

**Figure 5 ijms-26-01259-f005:**
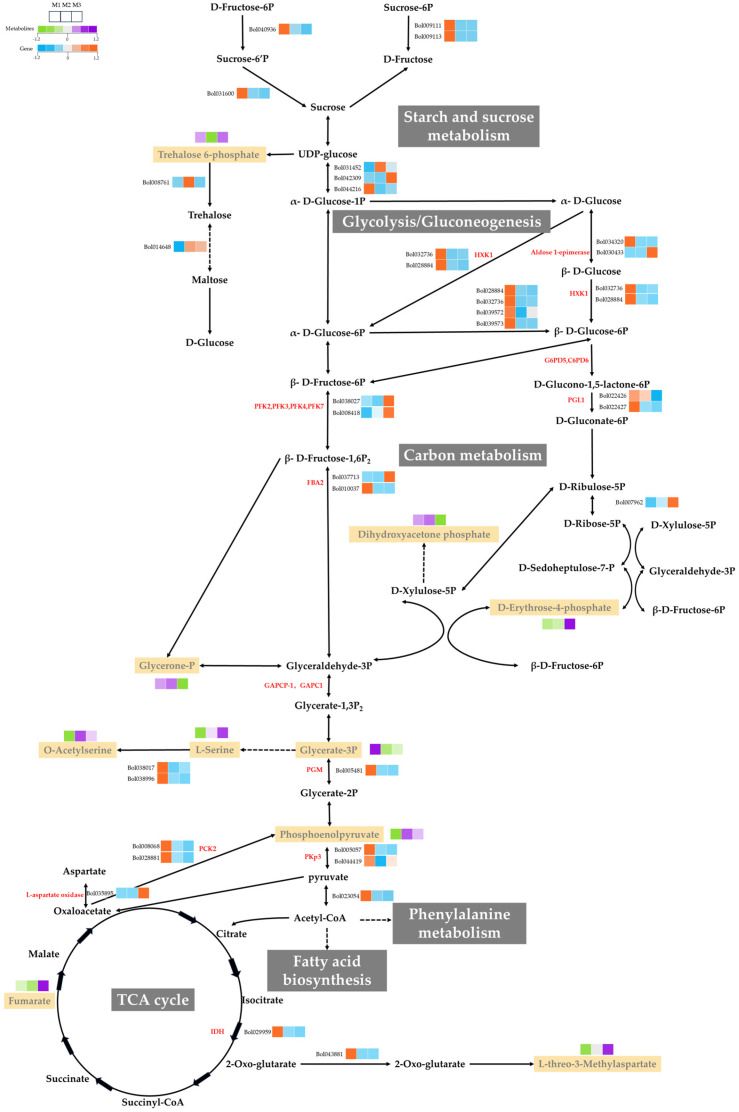
Carbohydrate metabolic pathway diagram. Yellow boxes highlight differential metabolites. The enzymes and proteins shown include HXK1 (hexokinase 1), G6PD5 (glucose-6-phosphate dehydrogenase 5), G6PD6 (glucose-6-phosphate dehydrogenase 6), PGL1 (6-phosphogluconolactonase 1), PFK3 (6-phosphofructokinase 3), PFK2 (6-phosphofructokinase 2), PFK7 (6-phosphofructokinase 7), PFK4 (6-phosphofructokinase 4), FBA1 (fructose–bisphosphate aldolase, class I), FBA2 (fructose–bisphosphate aldolase, class I), GAPCP-1 (glyceraldehyde-3-phosphate dehydrogenase), GAPC1 (glyceraldehyde-3-phosphate dehydrogenase, cytosolic), PGM (phosphoglycerate mutase-like protein), PKp3 (plastidial pyruvate kinase 3), PKP-BETA1 (pyruvate kinase BETA1), PCK2 (phosphoenolpyruvate carboxykinase 2), and IDH (isocitrate dehydrogenase).

**Figure 6 ijms-26-01259-f006:**
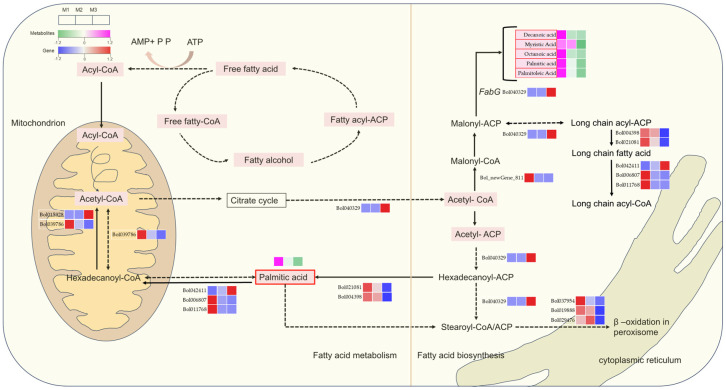
Regulatory network pathways of fatty acid synthesis and metabolism. Red boxes indicate differential metabolites. FabG: 3-Ketoacyl-ACP reductase.

**Figure 7 ijms-26-01259-f007:**
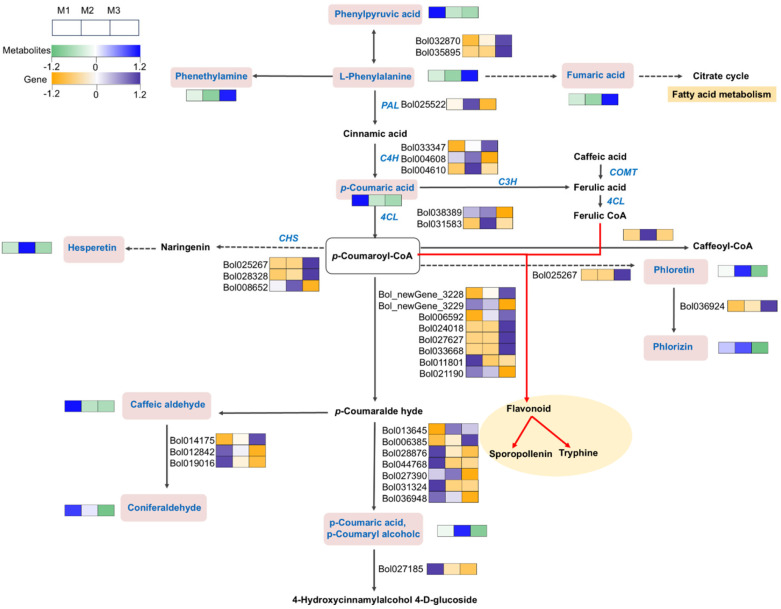
Network diagram of phenylpropanoid biosynthesis and metabolic pathway regulation. Pink boxes indicate differential metabolites, yellow boxes indicate metabolic pathways, and blue text indicates key enzymes in the pathway. PAL: cinnamic acid 4-hydroxylase; C4H: cinnamic acid 4-hydroxylase; 4CL: 4-coumarate-CoA ligase; CHS: chalcone synthase; COMT: caffeic acid O-methyltransferase; C3H: coumaric acid 3-hydroxylase.

**Figure 8 ijms-26-01259-f008:**
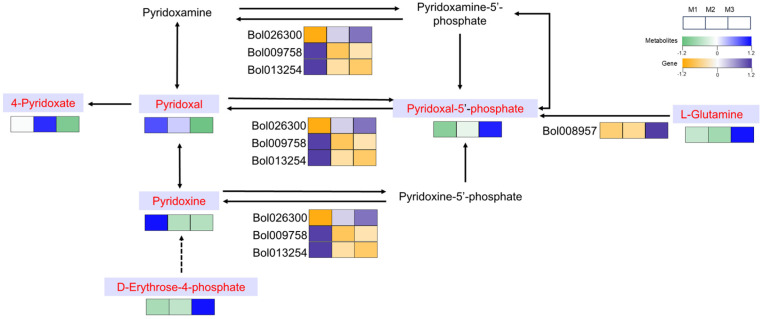
Vitamin B6 metabolic pathway. Purple boxes indicate differential metabolites.

**Figure 9 ijms-26-01259-f009:**
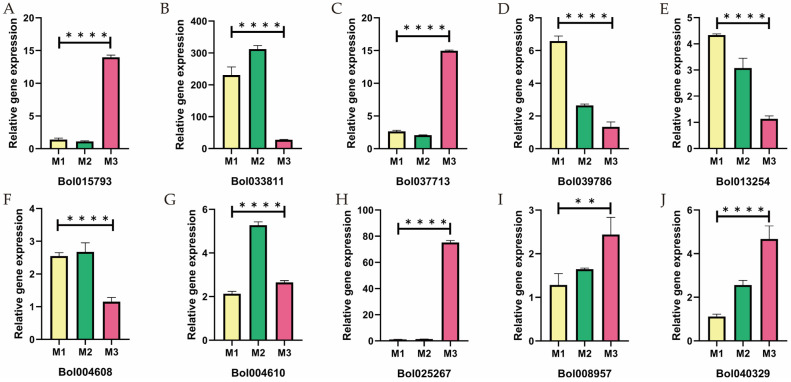
Genes associated with key regulatory networks verified using qRT-PCR. The significance levels are ** *p* < 0.05 and **** *p* < 0.001.

**Table 1 ijms-26-01259-t001:** Statistical table of the sequencing data.

Sample ID	Clean Reads	Clean Bases	GC (%)	≥Q30 (%)
M1-1	22073512	6588876216	47.27	93.62
M1-2	25223728	7547719730	46.88	91.61
M1-3	19246583	5759013734	46.93	92.26
M2-1	25705343	7665178646	47.55	94.13
M2-2	31271530	9327533842	47.39	94.21
M2-3	22920781	6856184676	47.25	93.19
M3-1	19767572	5914870538	47.31	92.97
M3-2	21603206	6462452108	47.32	92.81
M3-3	22816605	6826911388	47.42	93.14

Note: Sample IDs are as follows: M1 corresponds to 20-2-5, M2 corresponds to DGMS20-2-5, and M3 corresponds to Ogura20-2-5. The terms used in the analysis are defined as follows: clean reads: the total number of paired-end reads in the clean data; clean bases: the total number of bases in the clean data; GC: GC content; ≥Q30%: the percentage of bases in the clean data with a base call quality score greater than or equal to 30.

## Data Availability

All data supporting the findings of this study are available in the paper and its Appendix A published online.

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
