# Peer review of "Integrated Transcriptomics and Metabolomics Analysis Reveals Convergent and Divergent Key Molecular Networks of Dominant Genic Male Sterility and Cytoplasmic Male Sterility in Cabbage"

_ijms, 2025, doi:10.3390/ijms26031259_

Round 1
Reviewer 1 Report
Comments and Suggestions for Authors
In my opinion, this study on the comparison of different Brassica lines to study their fertility based on key genes related to pollen development is interesting. In general, I think the research is well done with conclusive results. Nevertheless, in my opinion, the text should be improved in some parts with an improvement of the background of the bibliographical references. Some details in material and methods should be included. Also, the statistics should be included in some parts of the analyses provided. Some additional details are included in the attached file.

In my opinion, some parts of the text should be rewritten to polish the text.
Reviewer 2 Report
Comments and Suggestions for Authors
The manuscript submitted by Zhang and coworkers describes a comparative analysis of gene expression and metabolomics in lines of Brassica oleracea with cytoplasmic and dominant genomic sterility. The analysis provides an initial molecular description of the two types of sterility and extensive datasets for further analysis. The careful work meets standard expectations set in the field, and the text is well-written. However, the presentation needs some improvements before publication.
1. Please provide a supplementary table of comparative gene expression data, similar to the metabolomic data in Table S1.
2. Please include a legend to Table S1.
3. Several figures need specifications of measurements and units on the ordinates.
4. The font size of some lettering on most figures is too small. There is plenty of room for more legible lettering.
5. Some legends include Chinese text that should be translated into English, the standard language of IJMS.
6. Provide definitions of all abbreviations at their first appearance.
